# Assisted Reproductive Technique and Abnormal Cord Insertion: A Systematic Review and Meta-Analysis

**DOI:** 10.3390/biomedicines10071722

**Published:** 2022-07-17

**Authors:** Shinya Matsuzaki, Yutaka Ueda, Satoko Matsuzaki, Yoshikazu Nagase, Mamoru Kakuda, Misooja Lee, Michihide Maeda, Hiroki Kurahashi, Harue Hayashida, Tsuyoshi Hisa, Seiji Mabuchi, Shoji Kamiura

**Affiliations:** 1Department of Gynecology, Osaka International Cancer Institute, Osaka 541-8567, Japan; michihide.maeda@oici.jp (M.M.); hiroki.kurahashi@oici.jp (H.K.); harue.hayashida@oici.jp (H.H.); tsuyoshi.hisa@oici.jp (T.H.); seiji.mabuchi@oici.jp (S.M.); kamiura-sh@oici.jp (S.K.); 2Department of Obstetrics and Gynecology, Osaka University Graduate School of Medicine, Osaka 565-0871, Japan; mamorukakuda@gmail.com; 3Department of Obstetrics and Gynecology, Osaka General Medical Center, Osaka 558-8558, Japan; satokooooo.0825@gmail.com; 4Department of Obstetrics and Gynecology, Kaizuka City Hospital, Osaka 597-0015, Japan; doctoryoshikazu@gmail.com; 5Department of Forensic Medicine, School of Medicine, Kindai University, Osaka 589-8511, Japan; misooja-88@umin.ac.jp

**Keywords:** assisted reproductive technique, velamentous cord insertion, marginal cord insertion, vasa previa, cesarean section, cesarean delivery, instrumental delivery, placental histopathology

## Abstract

Abnormal cord insertion (ACI) is associated with adverse obstetric outcomes; however, the relationship between ACI and assisted reproductive technology (ART) has not been examined in a meta-analysis. This study examines the association between ACI and ART, and delivery outcomes of women with ACI. A systematic review was conducted, and 16 studies (1990–2021) met the inclusion criteria. In the unadjusted pooled analysis (*n* = 10), ART was correlated with a higher rate of velamentous cord insertion (VCI) (odds ratio (OR) 2.14, 95% confidence interval (CI) 1.64–2.79), marginal cord insertion (*n* = 6; OR 1.58, 95%CI 1.26–1.99), and vasa previa (*n* = 1; OR 10.96, 95%CI 2.94–40.89). Nevertheless, the VCI rate was similar among the different ART types (blastocyst *versus* cleavage-stage transfer and frozen *v**ersus* fresh embryo transfer). Regarding the cesarean delivery (CD) rate, women with VCI were more likely to have elective (*n* = 3; OR 1.13, 95%CI 1.04–1.22) and emergent CD (*n* = 5; OR 1.93, 95%CI 1.82–2.03). In conclusion, ART may be correlated with an increased prevalence of ACI. However, most studies could not exclude confounding factors; thus, further studies are warranted to characterize ART as a risk factor for ACI. In women with ACI, elective and emergent CD rates are high.

## 1. Introduction

Abnormal cord insertion (ACI) is divided into two major groups: the marginal cord insertion (MCI) and velamentous cord insertion (VCI) groups [1]. VCI is characterized by membranous cord vessels at the cord insertion site [2] and its estimated prevalence has been reported to be approximately 1%. Several studies have shown that VCI is associated with increased adverse obstetric outcomes, such as preterm birth (PTB), increased rate of cesarean deliveries (CD), and small for gestational age (SGA) status [3,4]. Nevertheless, the results of previous studies that examined the association between MCI and obstetric outcomes were inconsistent [1,5,6,7]. Overall, contrary to VCI, the effect of MCI on the obstetric outcomes appears to be modest.

The possible reason behind the adverse effects of ACI, especially in VCI cases, on obstetric outcomes may be the lack of Wharton’s jelly around the cord vessels [8]. The association between the lack of Wharton’s jelly and clinically significant placental pathology has been reported, and unprotected vessels are prone to compression and rupture, thus increasing maternal and fetal morbidity [8]. Therefore, awareness of the risk factors for ACI, as well as the risk of adverse obstetric outcomes because of ACI, may be helpful in predicting poor maternal and fetal outcomes.

Recognizing the risk factors for ACI, including vasa previa, may be helpful in the antenatal diagnosis of patients with such conditions. However, the risk factors for ACI are understudied, especially in patients with vasa previa [9]. Although assisted reproductive technology (ART) has been reported as a risk factor for ACI, including vasa previa, the effect of ART on the prevalence of ACI has not been determined in a meta-analysis [10]. Notably, ART is associated with an increased prevalence of some placental disorders, such as placenta previa and placenta accreta spectrum disorder (PASD). Interestingly, recent systematic reviews have found that the risk of placenta previa and PASD is different among ART types [11]. For instance, a frozen embryo transfer is associated with a higher incidence of PASD than a fresh embryo transfer [11,12]. We consider that the risk of ACI may be different among different types of ART pregnancies. Such differences in the risk of ACI between ART types may help elucidate the mechanism underlying the development of ACI.

The variant of VCI that is associated with the highest amount of adverse obstetric outcome is vasa previa, which is characterized by unprotected cord vessels running within 2 cm of the internal cervical os [13,14,15], and the combination of low-lying placenta and VCI has been recognized as a high-risk condition of vasa previa [9]. The estimated prevalence of vasa previa has been reported to be approximately 0.05%, and undiagnosed vasa previa is associated with an extremely high fetal mortality rate, reaching up to 44% [16]. Thus, the antenatal diagnosis of vasa previa is essential to improve fetal morbidity and mortality. Previous studies have found that the prevalence of vasa previa appears to be high (approximately 0.3–0.5%) in women who conceived using ART [9,16,17,18,19]; however, comparative studies examining the rate of vasa previa between ART pregnancy and spontaneous pregnancy are scarce. The effect of VCI on the rate of low-lying placenta is also unclear.

Therefore, this study aimed to examine the relationship between ART and ACI. As an ACI may be associated with adverse delivery outcomes, the relationship between ACI and delivery outcomes was also examined.

## 2. Materials and Methods

### 2.1. Approach to the Systematic Literature Review

A systematic review (PROSPERO registration ID: CRD42021261424) was performed to determine the effect of ART on the prevalence of ACIs. The outcomes of interest were as follows: (i) the effect of ART on the prevalence of ACI, (ii) the effect of ART on the prevalence of ACI examined according to ART type, and (iii) the effect of ACI on delivery outcomes.

In accordance with the Preferred Reporting Items for Systematic Reviews and Meta-Analyses statement (version 2020) [20], a systematic literature search was conducted using the PubMed, Cochrane Central Register of Controlled Trials (CENTRAL), and Scopus databases. Literature published before 31 May 2021, was screened using words related to ART and ACI. The used search engines and search date were based on the pre-registered protocol in PROSPERO. Medical subject headings (MeSH terms) were used in the PubMed and Cochrane database searches.

### 2.2. Eligibility Criteria, Information Sources, and Search Strategy

Studies were screened by inspecting their titles and abstracts. All abstracts were screened by authors Sh. M. and M. L. to identify studies that examined the association between ART and ACI (Appendix A).

### 2.3. Study Selection

To determine the effect of ART on the prevalence of ACI (primary aim of this study), studies that met the following criteria were included: (1) comparative studies comparing experimental (ART pregnancy) and control groups (non-ART pregnancy); (2) ACI criteria comprising vasa previa in pregnant women who conceived via ART; and (3) clear identification of the number of women with ACI in the ART group. Among the eligible studies, comparative studies that examined the outcome of interest (secondary aim of this study or sensitivity analysis) by comparing the VCI and control groups were further determined.

The exclusion criteria were as follows: (1) insufficient information to clearly identify the number of patients with ACI; (2) articles not written in English; and (3) conference abstracts, reviews, systematic reviews, and meta-analyses.

### 2.4. Data Extraction

All information was extracted by the leading author (Sh. M.). The leading author’s name, publication year, study location, total number of cases, number of control and experimental groups, and outcomes of interest were recorded. The data included in the analysis were verified by the review author (M. L.).

### 2.5. Analysis of Outcome Measures and Assessment of Bias Risk

The primary objective of this study was to examine the effect of ART on the rate of ACIs. Since an ACI may be associated with adverse outcomes during delivery, the delivery outcomes of women with ACI compared with those without ACI were considered as secondary outcomes. In the sensitivity analysis, the effect of VCI on the rate of low-lying placentas was examined. As the specific analysis of low-lying placenta is lacking, the effect of VCI on the prevalence of abnormal placentation was examined. Another sensitivity analysis was performed to explore the patient background that affected the rate of elective and emergent CD. Another sensitivity analysis was performed, in which the studies that included only women with a twin pregnancy were excluded. Some analyses were added to the registered protocol to enhance the discussion of the relationship between ACI and delivery outcomes, ART, and vasa previa.

The risk of bias in the included studies was assessed using the risk of bias in non-randomized studies-of interventions tool, as previously performed [21,22,23].

### 2.6. Meta-Analysis Plan

Information was collected from eligible studies, and the hazard of the outcomes of interest was calculated using the 95% confidence intervals (CIs) of the values to calculate the odds ratios (ORs) for these outcomes. Heterogeneity among the included studies was determined using the *I*^2^ statistical test to quantify the percentage of total variation. A meta-analysis was also performed, and all graphics were constructed using the RevMan version 5.4.1 software (Cochrane Collaboration, Copenhagen, Denmark). Heterogeneity across studies was evaluated using the *I*^2^ value, and a fixed or random effect analysis was performed (Appendix A) [24]. In the created images, the size of the colored box characterizes the weight of study. A black horizontal line represents the 95% CI of the study result, with each end of the line representing the CI boundaries. The black diamond represents the combined results of the studies.

### 2.7. Statistical Analysis

Differences in baseline demographics between both groups were analyzed using Fisher’s exact test or the chi-squared test, as appropriate. All statistical analyses were based on two-sided hypotheses, and a *p*-value < 0.05 was considered statistically significant. The Statistical Package for Social Sciences version 28.0 (SPSS, IBM, Armonk, NY, USA) was used for all analyses.

## 3. Results

### 3.1. Study Selection

The study selection scheme is illustrated in Figure 1. Overall, 1463 studies were examined, and 16 studies (37,128 ART pregnancies and 1,597,784 non-ART pregnancies) met the inclusion criteria for the descriptive analysis [7,25,26,27,28,29,30,31,32,33,34,35,36,37,38,39]. As the two studies reported by Ebbing et al. [33,35] used overlapping data (same database), the outcome of interest was different between them. Therefore, the analysis of the effect of ART on the VCI rate [33] and the analysis of delivery outcomes and patient characteristics [35] were used in previous studies.

#### Risk of Bias of Eligible Studies

The risk of bias assessment for the included studies is presented in Appendix A. Of those (*n* = 16), a possible moderate publication bias (moderate quality) in 10 studies [7,29,30,31,33,34,35,36,37,38] and severe publication bias (low quality) in the other six studies [25,26,27,28,32,39] were observed.

### 3.2. Study Characteristics

The metadata of the 16 included studies are summarized in Appendix A. Among eligible studies (*n =* 16), the year of publication was between 1990 and 2021, and all studies were retrospective [7,25,26,27,28,29,30,31,32,33,34,35,36,37,38,39]. No prospective studies or randomized controlled trials have been identified. Two studies included only twin pregnancies [32,37]. Among the eligible studies (*n =* 16), approximately half were from Europe (*n* = 7, 43.8%) [32,33,35,36,37,38,39], followed by Japan (*n* = 3, 18.8%) [25,31,34], the United States (*n* = 3, 18.8%) [26,28,30], Canada (*n* = 2, 12.5%) [7,27], and China (*n* = 1, 6.3%) [29].

#### 3.2.1. Number of Studies: Primary Outcome

A systematic literature search was conducted to identify studies that examined the relationship between ART and ACI; 11 eligible studies were found [7,29,30,31,32,33,34,36,37,38,39]. Of these studies (*n* = 11), 10 examined the effect of ART and VCI [7,29,30,31,32,33,34,36,37,39], 6 examined the relationship between ART and MCI [7,30,32,33,37,39], and 1 examined the association between ART and vasa previa [38].

Four studies compared the rate of VCI among different types of ART [25,26,27,28]. Of these studies (some overlapped), three compared the rate of VCI between blastocyst transfer and cleavage transfer [25,27,28], two compared the effect of frozen embryo transfer (ET) *versus* fresh ET on the rate of VCI [25,28], and one examined the effect of preimplantation genetic testing (PGT) on the rate of VCI [26].

#### 3.2.2. Number of Studies: Secondary Outcomes

Six eligible studies were identified, and the number of studies that examined each outcome is listed below [7,25,29,34,35,36]. (i) The rate of CD (VCI, *n* = 4 and MCI, *n* = 2) [7,29,35,36], (ii) the rate of elective CD (VCI, *n* = 3 and MCI, *n* = 1) [29,35,36], (iii) the rate of emergent CD VCI, *n* = 5 and MCI, *n* = 1) [25,29,34,35,36], (iv) the instrumental delivery rate (VCI, *n* = 4 and MCI, *n* = 2) [7,25,35,36], and (v) the rate of postpartum hemorrhage have not been determined.

#### 3.2.3. Number of Studies: A Sensitivity Analysis

The following studies compared each patient’s background between women with and without VCI. Especially, they examined the following: (a) the presence of abnormal placentation by assessing (i) the rate of placenta previa (*n* = 4) [29,34,35,36] and (ii) the rate of PASD (*n* = 2) [25,34]; and (b) the patient background to estimate the indication for elective CD by assessing (i) the rate of nulliparous cases (*n* = 6) [7,25,29,33,34,36], (ii) the rate of prior CD (*n* = 3) [29,35,36], and (iii) the rate of fetal malpresentation (*n* = 1) [35]. No study has clarified the indications for emergent CD.

As for comparative studies that compared the delivery outcomes of women with and without VCI, none explored indications for CD, such as non-reassuring fetal status and arrest of labor.

### 3.3. Primary Outcome: Relationship between ART and ACI

#### 3.3.1. Effect of ART on VCI Incidence

A meta-analysis was conducted to determine the effect of ART on the VCI rate using 10 retrospective studies (Table 1). Owing to considerable heterogeneity, a random-effect analysis was performed. In the unadjusted pooled analysis (*n* = 10) [7,29,30,31,32,33,34,36,37,39], pregnancies by ART were associated with an increased rate of VCI compared to pregnancies without ART (Figure 2) (OR 2.14, 95%CI 1.64–2.79; heterogeneity, *p* < 0.01; and *I*^2^ = 75%). Three studies determined the effect of ART on VCI incidence using a multivariate analysis. In the adjusted pooled analysis, ART was correlated with a higher incidence of VCI (OR 2.11, 95%CI 1.91–2.33).

In the sensitivity analysis, studies that included only women with a twin pregnancy [32,37] were excluded. In this unadjusted pooled analysis (*n* = 8) [7,29,30,31,33,34,36,39], women with an ART pregnancy were more likely to have VCI compared to those with a non-ART pregnancy (OR 2.28, 95%CI 1.64–3.15; heterogeneity, *p* < 0.01; and *I*^2^ = 77%).

#### 3.3.2. Effect of ART on MCI Incidence

Six studies investigated the influence of ART on MCI prevalence [7,30,32,33,37,39]. As considerable heterogeneity was detected among studies, a random-effect analysis was used. In the unadjusted analysis (*n* = 6), ART pregnancy was associated with an increased MCI prevalence compared to non-ART pregnancies (Figure 2; OR 1.58, 95%CI 1.26–1.99; heterogeneity, *p* < 0.01; *I*^2^ = 78%). Two studies have examined the relationship between ART pregnancy and MCI incidence using a multivariate analysis (Table 1). In the adjusted pooled analysis (*n* = 2), a significant higher incidence of MCI was observed in women who conceived by ART compared to those who conceived by non-ART (OR 1.42, 95%CI 1.33–1.51; heterogeneity, *p* = 0.36; *I*^2^ = 0).

In the sensitivity analysis, studies with only twin pregnancies [32,37] were excluded. In the unadjusted pooled analysis, (*n* = 4) [7,30,33,39], ART pregnancies were correlated with a higher MCI rate compared to non-ART pregnancies (OR 1.74, 95%CI 1.49–2.04; heterogeneity, *p* < 0.01; *I*^2^ = 42%).

#### 3.3.3. Influence of ART on the Incidence of Vasa Previa

The effect of ART on the incidence of vasa previa was studied in only one report [38]. In this study, ART pregnancy was significantly associated with a higher rate of vasa previa (OR 10.96, 95%CI 2.94–40.89). No studies have performed a multivariate analysis to investigate the influence of ART on the prevalence of vasa previa.

### 3.4. Primary Outcome: Effect of Different Types of ART on the Rate of ACI

#### 3.4.1. Blastocyst Transfer versus Cleavage-Stage Transfer

Three comparator studies compared the effects of blastocyst transfer and those of cleavage-stage transfer on the rate of VCI, whereas no study compared MCI and vasa previa [25,27,28]. The results of the three studies are inconsistent; Furuya et al. showed that blastocyst transfer was associated with a higher rate of VCI compared to cleavage-stage transfer [25], whereas Volodarsky et al. reported the opposite results [27]. Moreover, in the study reported by Sacha et al., the VCI rate was not significantly different between blastocyst transfer and cleavage-stage transfer [28]. In the unadjusted pooled random effect analysis, there was no significant difference in the VCI rate between both transfers (Figure 3; OR 1.33, 95%CI 0.50–3.53; heterogeneity, *p* < 0.01; and *I*^2^ = 87%). The adjusted pooled analysis demonstrated results similar to those of the univariate analysis (OR 1.53, 95%CI 0.17–13.68; heterogeneity, *p* < 0.01; and *I*^2^ = 94%).

#### 3.4.2. Frozen ET Versus Fresh ET

The rate of VCI between frozen ET *versus* fresh ET was compared between two studies (Table 2) [25,28]. The rate of VCI was similar between frozen ET and fresh ET in both the unadjusted (*n* = 2; OR 1.22, 95%CI 0.79–1.89; heterogeneity, *p* = 0.36; *I*^2^ = 0%) and adjusted analyses (*n* = 1; OR 1.58, 95%CI 0.79–3.55).

#### 3.4.3. Effect of PGT on VCI

The effect of PGT on VCI frequency was determined in one study, which demonstrated a similar rate of VCI between women who conceived ART with and without PGT (*n* = 1; OR 2.52, 95%CI 0.77–8.20).

### 3.5. Secondary Outcomes: Delivery Outcomes of Women with ACI

#### 3.5.1. VCI Is Associated with an Increased Rate of Both Elective and Emergent CD

The influence of VCI on the rates of cesarean and instrumental deliveries was determined (Table 3 and Appendix A). Regarding the CD rate (*n* = 4), a fixed-effects analysis was used because there was no heterogeneity among studies. In the unadjusted pooled analysis, women with VCI were more likely to have CD (Figure 4; OR 1.65, 95%CI 1.57–1.73; heterogeneity, *p* = 0.89; *I*^2^ = 0%). Specific data regarding the rate of CD on women with VCI who conceived using ART were not available in the eligible studies. For further analysis, the effect of VCI on the rates of elective and emergent CD was determined.

To analyze the effect of VCI on elective and emergent CD, a fixed-effect analysis was conducted as no heterogeneity was observed among studies. In the unadjusted pooled analysis, VCI was associated with an increased rate of elective CD (*n* = 3; OR 1.13, 95%CI 1.04–1.22; heterogeneity, *p* = 0.55; *I*^2^ = 0%). Similarly, in the examination of emergent CD, women with VCI had a significantly higher CD rate than those without VCI (*n* = 5; OR 1.93, 95%CI 1.82–2.03; heterogeneity, *p* = 0.68; and *I*^2^ = 0%).

#### 3.5.2. MCI Has a Potential to Be Associated with an Increased Rate of CD

Similar to the analysis of VCI, the effect of MCI was determined according to the rate of CD (Appendix A). Two studies examined the effects of MCI (Appendix A); however, their results were conflicting. O’Quinn et al. found that the rate of CD was similar between women with and without MCI (OR 1.01, 95% CI 0.89–1.14) [7], whereas Ebbing et al. revealed that MCI was correlated with a higher rate of CD (OR 1.37, 95% CI 1.33–1.40) [35]. In the unadjusted pooled fixed-effects analysis, the CD rate was similar between women with and without MCI (*n* = 2; OR 1.18, 95% CI 0.88–1.59; heterogeneity, *p* < 0.001; *I*^2^ = 96%).

Only one study stratified the rate of CD into elective and emergent CD (Appendix A) and found that MCI was associated with an increased rate of both types (elective: OR 1.21, 95% CI 1.16–1.26; and emergent: OR 1.42, 95% CI 1.37–1.47) [35].

#### 3.5.3. Instrumental Delivery Was Not Increased in Women with ACI

The influence of VCI on the rate of instrumental delivery was determined in four studies (Table 3) [7,25,35,36]. In the unadjusted fixed-effect analysis, the instrumental delivery rate was similar between women with and without VCI (Figure 4; *n* = 4; OR 1.00; 95% CI 0.93–1.07; heterogeneity, *p* = 0.78; *I*^2^ = 0%). Similarly, an unadjusted pooled analysis showed that MCI was not associated with an increased rate of instrumental delivery compared to pregnancies without MCI (*n* = 2; OR 1.00, 95% CI 0.96–1.03; heterogeneity, *p* = 0.98; *I*^2^ = 0%) [7,35].

### 3.6. Sensitivity Analysis: Patient Background and Obstetric Complications

#### 3.6.1. Abnormal Placentation

Specific data on low-lying placenta were not available in the eligible studies. Four studies examined the effects of VCI on the rate of placenta previa (Table 4) [29,34,35,36]. As no heterogeneity was observed among studies, a fixed-effects model was used. In the pooled unadjusted analysis, women with VCI were more likely to have placenta previa (*n* = 4; OR 3.57, 95% CI 2.98–4.28; heterogeneity, *p* = 0.45; and *I*^2^ = 0%) than those without VCI (Figure 5), and similar results were observed in the adjusted analysis (Figure 5; *n* = 4; OR 3.69, 95% CI 2.98–4.58; heterogeneity, *p* = 0.23; *I*^2^ = 30%). In the unadjusted analysis of the effect of VCI on PASD, patients with VCI were more likely to have PASD (*n* = 2; OR 3.60, 95% CI 1.61–8.05; heterogeneity, *p* = 0.75; *I*^2^ = 0%) than those without VCI [25,34]. Specific data on ART pregnancies with VCI were not available in the eligible studies.

#### 3.6.2. Patient Characteristics

We aimed to explore the reason behind the higher rate of CD in women with VCI compared to women without VCI (Table 5). Regarding age, there was no significant difference between women with and without VCI (*n* = 5; mean difference, 0.22; 95% CI, −0.17–0.62; heterogeneity, *p* = 0.14; and *I*^2^ = 43%) in the pooled random-effect analysis (Figure 6) [7,25,29,32,36]. In the sensitivity analysis, the study that included only twin pregnancy was removed [32], and fixed effect analysis was performed. The results of sensitivity analysis were similar with those of primary analysis (*n* = 4; mean difference 0.07, 95% CI −0.18–0.32; heterogeneity *p* = 0.59, and *I*^2^ = 20%).

The rate of nulliparity was compared between women with and without VCI using a random effects analysis. In this pooled analysis, women with VCI were more likely to be nulliparous than those without VCI (*n* = 6; OR 1.34, 95%CI 1.18–1.52; heterogeneity, *p* = 0.03; and *I*^2^ = 60%) [7,25,29,33,34,36].

#### 3.6.3. Indications for Elective CD

No studies have clarified an indication for elective CD through a comparison of women with and without VCI. As the major indications for elective CD except for abnormal placentation are prior CD, fetal malpresentation, and multiple gestation, these factors were explored. The rates of prior CD and fetal malpresentation were determined to assess the higher rate of elective CD in women with VCI compared to those without. Because of moderate heterogeneity among studies, a random-effect analysis was conducted. A similar rate of prior CD was observed between women with and without VCI (*n* = 3; OR 1.00, 95%CI 0.82–1.22; heterogeneity; *p* = 0.11; and *I*^2^ = 54%) [29,35,36]. Only one study compared the rate of fetal malpresentation between women with and without VCI. Ebbing et al. reported a significantly higher prevalence of fetal malpresentation in the former type of women than in the latter (*n* = 1; OR 2.19, 95%CI 2.03–2.36) [35]. The rate of CD for multiple gestation was not available in the included studies.

#### 3.6.4. Indications for Emergent CD

No studies have explored the indications for emergent CD, such as a non-reassuring fetal status and arrest of labor in women with ACI.

## 4. Discussion

### 4.1. Key Findings

The key findings of this study are as follows: (i) ART is associated with an increased prevalence of ACI, including vasa previa; (ii) ACI is associated with an increased rate of both elective and emergent CD, but not with the rate of instrumental delivery; (iii) women with VCI may be more likely to have abnormal placentation compared to those without VCI; and (iv) the mechanism by which ART causes the development of VCI has not been reported in basic research. While the meta-analysis of the relationship between ART and a higher prevalence of VCI is unique, the mechanism underlying the increased rate of ART-induced VCI has not been fully elucidated.

### 4.2. Comparison with Existing Literature

#### 4.2.1. Association between ART and ACI

In the past decades, previous studies have found that ART is associated with an increased rate of MCI and VCI [17,31,39,41,42]; however, these associations were not determined by a meta-analysis [4,43,44]. Yanaihara et al. compared the size of the placenta and umbilical cord between women with natural (*n* = 1453) and ART pregnancies (*n* = 157) [31]. There was no difference in the size and weight of the placenta or the length of the umbilical cord between these types of pregnancies. Jauniaux et al. reviewed the pathological features of the placenta by comparing spontaneous (*n* = 50) and ART pregnancies (*n* = 50). The authors found that ART pregnancy was associated with an increased rate of VCI and MCI, whereas the placental weight was not significantly different between the two groups [39].

Based on previous findings, the following two proposed hypotheses may aid in elucidating the mechanism underlying VCI development: the trophotropism hypothesis and the polarity hypothesis [42]. The trophotropism hypothesis is often used to explain VCI development, according to which the placenta in early pregnancy migrates with proceeding gestational weeks to a more vascularized area, so that its blood supply becomes better [45,46,47]. In the polarity hypothesis, the mechanism underlying the development of ACI is explained as follows: when the embryo does not contact the implantation base (blastocyst malrotation at implantation), the umbilical vessels need to spread between the placenta and the umbilical cord insertion site to reach the endometrium; thus, the placenta can develop appropriately in the endometrium, and the spread of umbilical cord vessels may lead to ACI [41,42,48,49].

In the meta-analysis of the current study, ART was associated with an increased rate of ACI, supporting the trophotropism theory. As the precise chronological sequence of biological actions required for blastocyst implantation is disrupted at multiple stages of ART, it may lead to an increase in the rate of ACI [42,49]. As the prevalence of ART pregnancy is increasing [50,51,52] and ACI is associated with adverse obstetric outcomes, the mechanism underlying ART-induced ACI needs to be elucidated. We believe that such studies have the potential to reduce ART-induced ACI.

While ART is widely recognized as a risk factor of vasa previa [9,17,18,19,53], our systematic review revealed that only one comparative study compared the rate of vasa previa between ART and non-ART pregnancies. Although a recent retrospective study was not included in this systematic review owing to the out-of-search date, this study compared the patient background between women with vasa previa and without vasa previa who were detected on the 20-week anomaly scan [54]. We calculated the odds ratio of ART on the rate of vasa previa using the results of this study, and we found that an ART pregnancy was associated with an increased rate of vasa previa compared to those with a non-ART pregnancy (OR 3.85, 95%CI 1.40–10.56). As the available studies are scant, further studies are warranted to examine the effect of ART on the prevalence of vasa previa.

One of the main combinations that cause vasa previa (type I) is VCI with a low-lying placenta (Figure 7A) [9,13,14]. Our study found that women with VCI were more likely to have a placenta previa than those without. A previous meta-analysis reported that ART pregnancies were associated with a higher rate of placenta previa than non-ART pregnancies (OR 2.96, 95%CI 2.43–3.60) [11]. Nevertheless, only one study showed a significantly higher incidence of low-lying placenta in ART pregnancy compared to that in non-ART pregnancy (adjusted OR 1.83, 95%CI 1.04–3.12) [55].

Although specific data regarding low-lying placenta is scant, we hypothesized that both women with VCI and women who conceived by ART have a potential to have a higher rate of low-lying placenta than women who conceived normally (Figure 7B). Based on the results of these meta-analyses, a possible mechanism for the increased rate of type I vasa previa in ART pregnancies is presented in Figure 7.

Our meta-analysis did not examine the relationship between ART and placental anomalies, including a bilobed placenta, a succenturiate lobe, and so on; thus, such an investigation may help further elucidate the mechanism behind type II vasa previa.

#### 4.2.2. Risk of ACI According to ART Type

In the meta-analysis, ART type (blastocyst transfer *versus* cleaved-stage transfer, frozen ET *versus* fresh ET, and with *versus* without PGT) did not correlate with an increased rate of VCI. Nevertheless, it is important to recognize that only a limited number of studies have examined the effect of VCI according to the ART type; therefore, this analysis is underpowered. Notably, the results of three studies that compared the rate of VCI between blastocyst transfer and cleavage-stage transfer are inconsistent. This discrepancy may be attributed to the following: (i) the different rates of frozen ET and fresh ET, (ii) the difference in endometrial thickness, and (iii) the difference in the included populations (age, nulliparity, race, and so on). To compensate for the limited number of examinations, a prospective or retrospective study that focuses on the rate of VCI among different types of ART, including a large number of women with matched patient backgrounds, is warranted.

#### 4.2.3. Effect of ACI on the Rate of CD

No previous studies have focused on the increased prevalence of elective CD in women with VCI. Therefore, the results of this study may be interesting; however, no studies have performed a multivariate analysis to examine the association between VCI and the rate of elective CD, to exclude the effect of confounding factors. Generally, the main indications for elective CD are prior CD, fetal malpresentation, and multiple gestations [56,57]; thus, confounding factors may have affected this increased rate of CD. We believe that a patient characteristics-adjusted analysis for the examination of the association between VCI and the rate of elective CD is warranted.

Previous studies have found that VCI is associated with an increased rate of emergent CD. While the included studies did not examine the reason for the increased rate of elective and emergent CD, women with VCI often have intrapartum abnormal fetal rates, especially variable decelerations, all of which may increase the rate of emergent CD [58]. The lack of Wharton’s jelly in umbilical cord vessels indicates unprotected cord vessels from compression during labor. This condition causes the occurrence of multiple variable decelerations, and these abnormal fetal heart tracings lead to increased emergent CD [59]. Notably, Hasegawa et al. reported that VCI at the lower third of the uterus is associated with increased abnormal intrapartum fetal heart tracings [60].

While the relationship between VCI and increased emergent CD has become robust, risk classification according to the emergent CD rate for VCI is still understudied. Moreover, although only a limited study on the following relationship is available, MCI has the potential to be associated with an increased rate of emergent CD [35]. We believe that future studies that classify the type of ACI according to emergent CD will be useful.

#### 4.2.4. Relationship between VCI and Abnormal Placentation

To the best of our knowledge, this study is the first to conduct a meta-analysis to determine whether women with VCI are more likely to have a placenta previa than those without VCI. To discuss the relationship between VCI and placenta previa, the following hypothesis proposed by Hasegawa is essential [49]: (i) the oblique location of the umbilical cord toward the lower segment of the uterus, resulting in the development of the placenta adjacent to the umbilical cord, may be associated with placenta previa; (ii) because of the atrophic changes in the placenta resulting from inadequate blood supply from the lower uterine wall, the unprotected abnormal vessels are longer in women with cord insertion in the lower uterine segment than in those with cord insertion in the middle to upper uterine segment. The results of this study support Hasegawa’s hypothesis.

### 4.3. Strengths and Limitations

To our knowledge, this is the first study to focus on the influence of ART on incidence of ACI, as well as on the effect of VCI on delivery outcomes. We found that ART pregnancy was associated with an increased rate of ACI compared to non-ART pregnancy. We also revealed that women with ACI showed higher rates of both elective and emergent CD, but not a higher rate of instrumental delivery. We believe that these results may help clinicians manage women with ACI.

Nevertheless, this study had several limitations. First, all included studies were retrospective; thus, unmeasured bias, such as selection bias, may exist. Other possible concerns in the included studies were as follows: (i) the quality of diagnosis and the definition of ACI may vary among studies; (ii) the indications for elective and emergent CD, and instrumental delivery may vary across studies; and (iii) the number of included studies was limited, in that it was not suitable to examine several outcomes of interest. These factors may have caused considerable bias in our study, as well as the presence of substantial heterogeneity among the eligible studies. Accordingly, these factors need to be recognized as notable limitations of this study.

Second, the effect of ART on the prevalence of VCI could not be determined by excluding confounding factors. Women conceived by ART may be more likely to be of advanced maternal age and have multiple gestation; thus, excluding confounding factors is essential. However, most studies [7,29,30,32,34,36,39] have not performed a multivariate analysis with adjustments for patient background; thus, our analysis cannot characterize ART as a risk factor for ACI.

Third, while exploring for indications of CD, although various obstetrics factors, including advanced maternal age, higher rate of nulliparous, placenta previa, prior CD, fetal malpresentation, and so on, are associated with an increased rate of CD, all studies have not performed a multivariate analysis with adjustments for these obstetric factors; thus, our analysis cannot illustrate VCI as a risk factor for CD by excluding confounding factors. Moreover, no studies clarified whether prior CD, fetal malpresentation, and multiple gestation are indications for elective CD. These limitations should be considered when interpreting the results of this study.

Fourth, while ART is associated with an increased rate of ACI, the underlying mechanism is still unknown because of the absence of basic research studies. Future studies examining this mechanism are warranted in order to enhance the robustness of the results of the current study. Fifth, to explore the relationship between VCI and adverse delivery outcomes, some outcomes of interest that were not listed in the registered protocol were determined, possibly causing bias in the systematic review. This also needs to be recognized as a limitation of this study.

Finally, only a limited number of studies have examined the effect of MCI on delivery outcomes. Moreover, MCI severity (umbilical cord insertion site according to the placental margin) was not determined in any study. The length of the umbilical cord insertion site from the placental margin and the effect of MCI need to be examined. Taken together, these notable limitations should be considered when interpreting the results of this study.

## 5. Conclusions

### 5.1. Implications for Practice

Although ACI is associated with an increased rate of CD, particularly emergent CD, the cause of the increased rate of emergent CD remains unclear. Elucidating the corresponding causes may contribute to a reduction in the rate of CD. Notably, it remains unclear which type of MCI and VCI is associated with a higher risk of CD (e.g., umbilical cord insertion to the lower uterine segment *versus* upper uterine segment and the distance from the cord insertion to the placental margin). These examinations may help clinicians manage women with ACIs during labor.

### 5.2. Implications for Clinical Research

In the current meta-analysis, ART may be associated with an increased rate of ACI, and ACI was correlated with an increased rate of elective and emergent CD. As the number of included studies is still limited and since most previous studies could not exclude confounding factors, extreme caution is needed when the results of our study are extrapolated. Therefore, future studies are warranted in order to examine these outcomes using a multivariate analysis and resolve the concerns regarding the cofounding factor in the analysis of the relationship between ART and ACI. We believe that a prospective study examining these relationships will also be useful.

Basic research that examines the relationship between ART and VCI may be of interest to prevent ART-induced VCI. Elucidating these mechanisms also has the potential to reveal the mechanisms underlying the development of VCI in spontaneous pregnancy. 

## Figures and Tables

**Figure 1 biomedicines-10-01722-f001:**
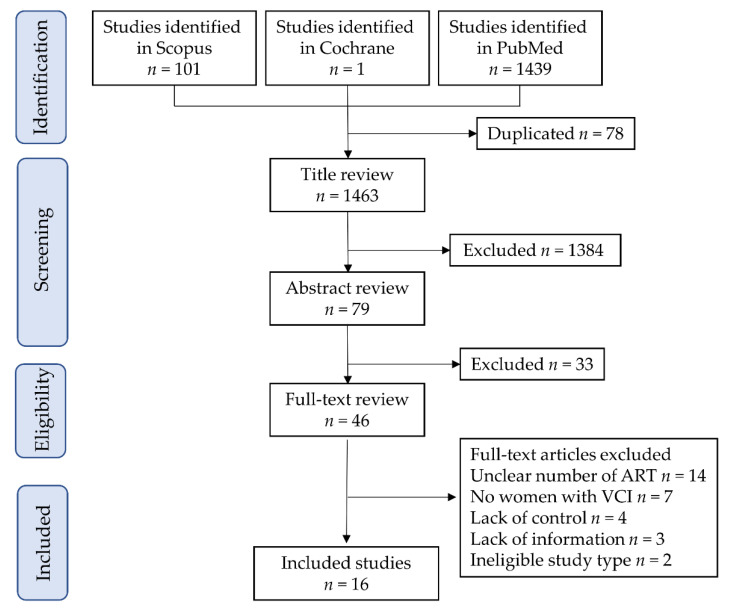
Study selection scheme of the systematic literature search. Abbreviations: ART, assisted reproductive technology; and VCI, velamentous cord insertion.

**Figure 2 biomedicines-10-01722-f002:**
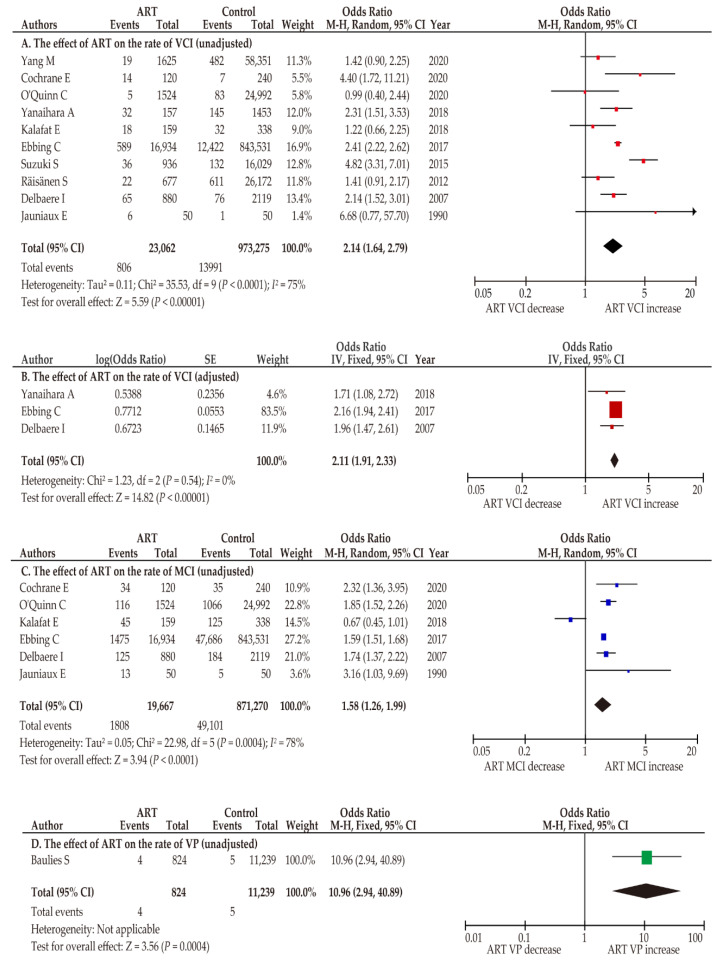
Effect of ART on the prevalence of ACI. Pooled odds ratios (ORs) were calculated using RevMan version 5.4.1 (Cochrane Collaboration, Copenhagen, Denmark); thus, some values may be slightly different from their original state. The position of the colored box is a point of the estimated OR. The effects of ART on the following conditions were determined through unadjusted or adjusted analyses: (**A**) VCI, (**B**) VCI (adjusted), (**C**) MCI, and (**D**) VP. Forest plots were ordered according to the year of publication. Substantial heterogeneity was observed in the unadjusted analyses ((**A**), *I*^2^ = 75%; (**C**), *I*^2^ = 78%) whereas no heterogeneity was observed in the adjusted analysis ((**B**), *I*^2^ = 0%). Abbreviations: ACI, abnormal cord insertion; VCI, velamentous cord insertion; MCI, marginal cord insertion; VP, vasa previa; ART, assisted reproductive technology; OR, odds ratio; CI, confidence interval; SE, standard error.

**Figure 3 biomedicines-10-01722-f003:**
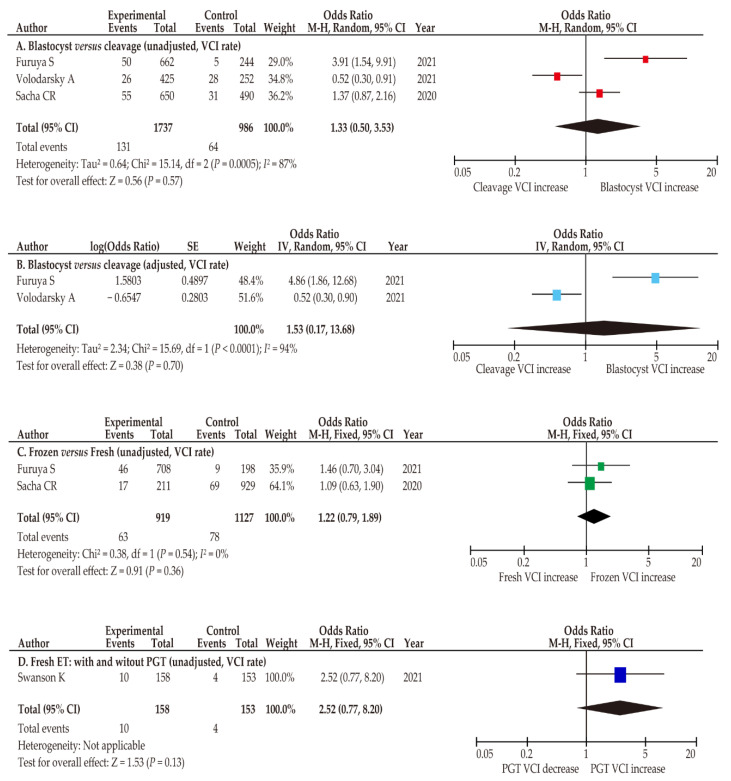
Risk of VCI according to ART type. Pooled odds ratios (ORs) were calculated using RevMan version 5.4.1 (Cochrane Collaboration, Copenhagen, Denmark). Thus, some values may be slightly different from their original state. The following comparisons were performed to determine VCI risk: (**A**,**B**) blastocyst transfer *versus* cleavage stage transfer (**A**, unadjusted; **B**, adjusted), (**C**) frozen ET *versus* fresh ET, and (**D**) frozen ET with *versus* without PGT. Forest plots were arranged according to the year of publication. Substantial heterogeneity was observed in the analysis of (**A**,**B**) (**A**, *I*^2^ = 87%; **B**, *I*^2^ = 94%), whereas no heterogeneity was observed in analysis of (**C**) (**C**, *I*^2^ = 0%). Abbreviations: VCI, velamentous cord insertion; ART, assisted reproductive technology; blastocyst, blastocyst transfer; cleavage stage transfer; frozen ET, frozen embryo transfer; fresh ET, fresh embryo transfer; PGT, preimplantation genetic testing; OR, odds ratio; CI, confidence interval; SE, standard error.

**Figure 4 biomedicines-10-01722-f004:**
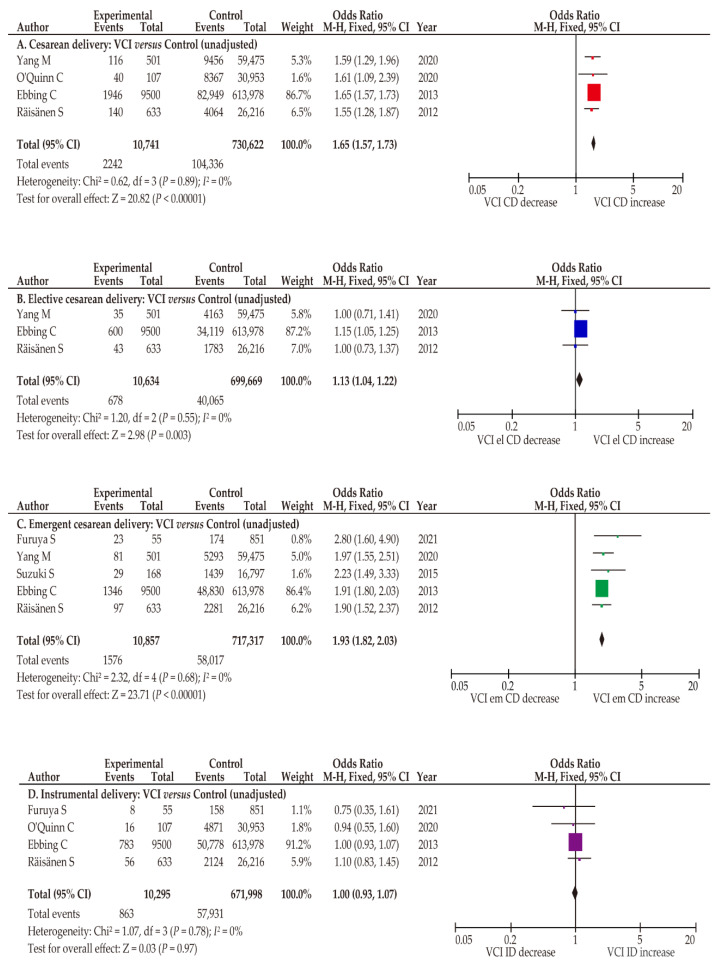
Effect of VCI on the rate of CD. Pooled odds ratios (ORs) were calculated using RevMan version 5.4.1 (Cochrane Collaboration, Copenhagen, Denmark). Thus, some values may be slightly different from their original state. The effect of VCI on the rate of CD was examined. No heterogeneity was observed during the unadjusted analysis (**A**–**D**, *I*^2^ = 0%), and fixed-effect analyses were performed. (**A**) Comparison of CD rates between women with and without VCI, (**B**) comparison of elective CD rates between women with and without VCI, (**C**) comparison of emergent CD rates between women with and without VCI, and (**D**) comparison of instrumental delivery rates between women with and without VCI. Abbreviations: VCI, velamentous cord insertion; ART, assisted reproductive technology; CD, cesarean delivery; el CD, elective cesarean delivery; em CD, emergent cesarean delivery; OR, odds ratio; CI, confidence interval; SE, standard error.

**Figure 5 biomedicines-10-01722-f005:**
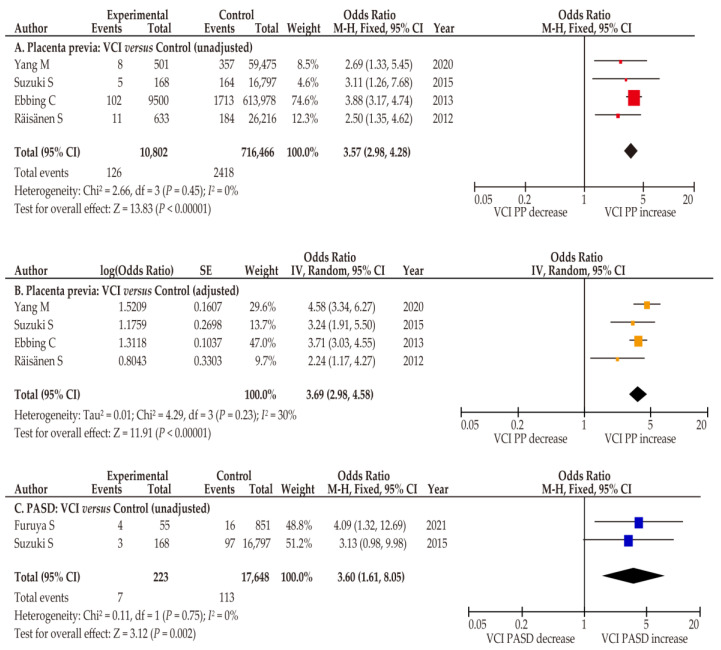
Effect of VCI on the prevalence of abnormal placentation. Pooled odds ratios (ORs) were calculated using RevMan version 5.4.1 (Cochrane Collaboration, Copenhagen, Denmark). Thus, some values may be slightly different from the original values. The effect of VCI on the incidence of abnormal placentation was determined. No heterogeneity was observed during the unadjusted analysis ((**A**,**C**) *I*^2^ = 0%), and fixed-effects analyses were performed. Moderate heterogeneity was observed in the adjusted analysis ((**B**), *I*^2^ = 30%), and a random-effect analysis was conducted. (**A**,**B**) Comparison of rates of placenta previa between women with and without VCI; (**C**) comparison of the prevalence of PASD between women with and without VCI. Abbreviations: VCI, velamentous cord insertion; PP, placenta previa; PASD, placenta accreta spectrum disorder; OR, odds ratio; CI, confidence interval; SE, standard error.

**Figure 6 biomedicines-10-01722-f006:**
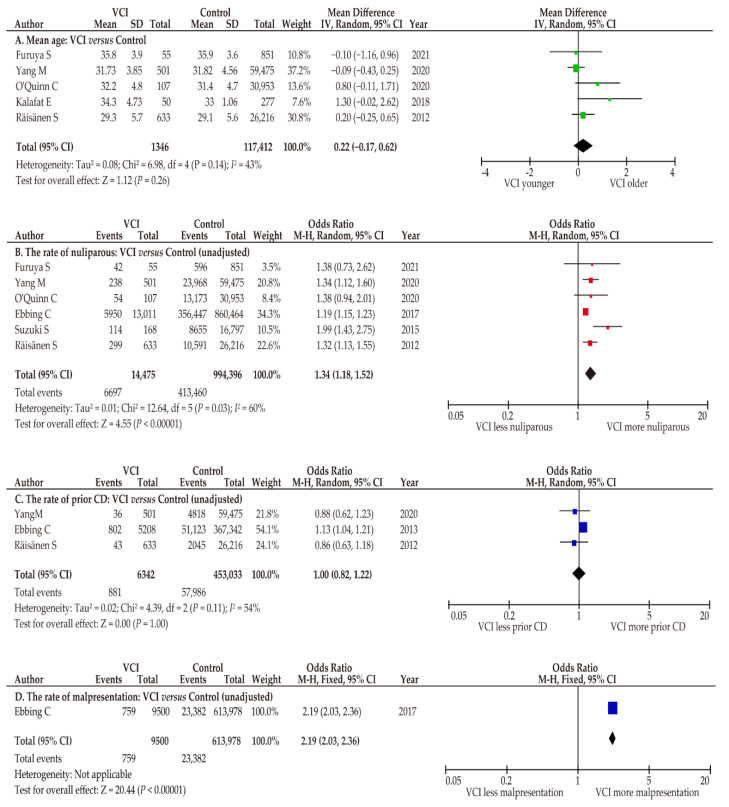
Differences in patient characteristics between women with and without VCI. Pooled ORs were calculated using RevMan version 5.4.1. (Cochrane Collaboration, Copenhagen, Denmark). Thus, some values may be slightly different from their original state. Because of moderate heterogeneity (**A**, *I*^2^ = 43%; **C**, *I*^2^ = 54%) and substantial heterogeneity (**B**, *I*^2^ = 60%), random-effect analyses were applied. (**A**) Mean age; (**B**) rate of nulliparity; (**C**) rate of prior CD; and (**D**) rate of fetal malpresentation. Abbreviations: VCI, velamentous cord insertion; CD, cesarean delivery; OR, odds ratio; CI, confidence interval; SE, standard error.

**Figure 7 biomedicines-10-01722-f007:**
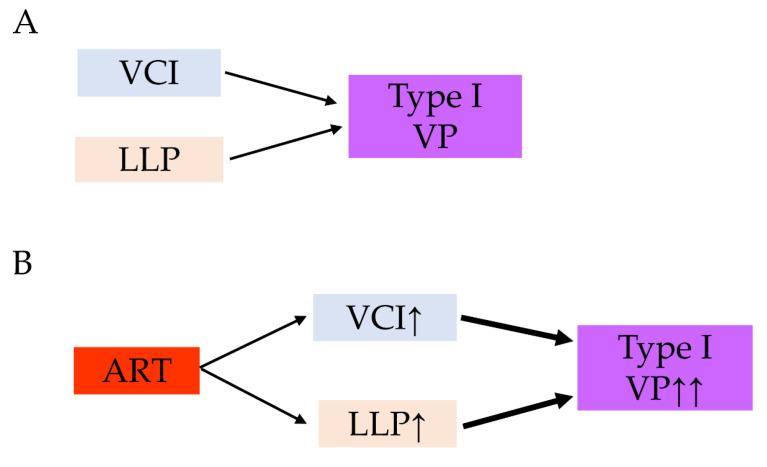
Our hypothesis behind the possible mechanism of the increased rate of vasa previa in ART pregnancies. (**A**) The combination of VCI and LLP is a high-risk condition of vasa previa. (**B**) ART pregnancy is associated with increased rate of VCI and LLP compared to non-ART pregnancy. The increased rate of VCI and LLP may lead to increased rate of type I vasa previa. Abbreviations: ART, assisted reproductive technology; VCI, velamentous cord insertion; VP, vasa previa; LLP, low-lying placenta; ↑, increase; ↑↑, markedly increase.

**Table 1 biomedicines-10-01722-t001:** The association between ART and ACI.

Author	Year	ART No.	ART ACI	Cont No.	Cont ACI	Crude OR (95%CI)	Adjusted OR (95%CI)
VCI							
Yang M [29]	2020	1625	19	58,351	482	1.41 (0.89–2.23)	--
Cochrane E [30]	2020	120	14	240	7	4.40 (1.72–11.21)	--
O’Quinn C [7]	2020	1524	5	24,992	83	0.99 (0.40–2.44)	--
Yanaihara A ^#^ [31]	2018	157	32	1453	145	2.31 (1.51–3.53)	1.72 (1.08–2.72)
Kalafat E * [32]	2018	159	18	338	32	1.22 (0.66–2.25)	--
Ebbing C [33]	2017	16,934	589	843,531	12,422	2.41 (2.22–2.62)	2.16 (1.94–2.41)
Suzuki S [34]	2015	936	36	16,029	132	4.82 (3.31–7.01)	--
Räisänen S [36]	2012	677	22	26,172	611	1.41 (0.91–2.17)	--
Delbaere I * [37]	2007	880	65	2119	76	2.14 (1.52–3.01)	1.96 (1.47–2.61)
Jauniaux E [39]	1990	50	6	50	1	6.68 (0.77–57.70)	--
MCI							
Cochrane E [30]	2020	120	34	240	35	2.32 (1.36–3.95)	--
O’Quinn C [7]	2020	1524	116	24,992	1066	1.85 (1.52–2.26)	--
Kalafat E * [32]	2018	159	45	338	125	0.67 (0.45–1.01)	--
Ebbing C [33]	2017	16,934	1475	843,531	47,686	1.59 (1.51–1.68)	1.43 (1.34–1.53)
Delbaere I * [37]	2007	880	125	2119	184	1.74 (1.37–2.22)	1.29 (1.05–1.59)
Jauniaux E [39]	1990	50	13	50	5	3.16 (1.03–9.69)	--
VP							
Baulies S [38]	2007	824	4	11,239	5	10.96 (2.94–40.89)	--

* All cases were twin pregnancies. ^#^ All embryos were frozen and transferred on day 5 (blastocyst stage). We calculated ORs using RevMan version 5.4.1 (Cochrane Collaboration, Copenhagen, Denmark), and some values were inferred by the authors; thus, certain values in the table were slightly different from the original values. Abbreviations: No., number of included cases; ART, assisted reproductive technology; ACI, abnormal cord insertion; ART ACI, the number of women with ACI in ART group; Cont ACI, the number of women with ACI in control group; VCI, velamentous cord insertion; MCI, marginal cord insertion; VP, vasa previa; Cont No., number of women in control group; OR, odds ratio; CI, confidence interval; --, not applicable.

**Table 2 biomedicines-10-01722-t002:** Risk of VCI according to ART type.

Author	Year	Exp No.	Exp Events	Cont No.	Cont Events	Crude OR (95%CI)	Adjusted OR (95%CI)
Blastocyst *versus* Cleavage							
Furuya S [25]	2021	662	50	244	5	3.91 (1.54–9.91)	4.86 (1.86–12.68)
Volodarsky A [27]	2021	425	26	252	28	0.52 (0.30–0.91)	0.5 (0.3–0.9)
Sacha CR [28]	2020	650	55	490	31	1.37 (0.87–2.16)	--
Frozen ET *versus* Fresh ET							
Furuya S [25]	2021	708	46	198	9	1.46 (0.70–3.04)	1.58 (0.79–3.55)
Sacha CR [28]	2020	211	17	929	69	1.09 (0.63–1.90)	--
Frozen ET with *versus* without PGT							
Swanson K [26]	2021	158	10	153	4	2.52 (0.77–8.20)	--

We calculated ORs using RevMan version 5.4.1 (Cochrane Collaboration, Copenhagen, Denmark), and some values were inferred by the authors; thus, certain numbers in the table may be slightly different from the original values. Abbreviations: ART, assisted reproductive technology; VCI, velamentous cord insertion; Exp No., number of experimental groups; Exp events, number of events in experimental group; Cont number, number of women in the control group; Cont events, number of events in control group; Crude OR, crude odds ratio; Adjusted OR, adjusted odds ratio; CI, confidence interval; Blastocyst, blastocyst transfer; Cleavage, cleavage-stage transfer; Frozen ET, frozen embryo transfer; Fresh ET, fresh embryo transfer; PGT, preimplantation genetic testing; --, not applicable.

**Table 3 biomedicines-10-01722-t003:** Effect of VCI on delivery outcomes.

Author	Year	Control No.	Control Events	VCI No.	VCI Events	Crude OR (95%CI)	Adjusted OR (95%CI)
CD (all)							
Yang M [29]	2020	59,475	9456	501	116	1.59 (1.29–1.96)	--
O’Quinn C [7]	2020	30,953	8367	107	40	1.61 (1.09–2.39)	--
Ebbing C [35]	2013	613,978	82,949	9500	1946	1.65 (1.57–1.73)	1.54 (1.47–1.62)
Räisänen S [36]	2012	26,216	4064	633	140	1.55 (1.28–1.87)	--
Elective CD							
Yang M [29]	2020	59,475	4163	501	35	1.00 (0.71–1.41)	--
Ebbing C [35]	2013	613,978	34,119	9500	600	1.15 (1.05–1.25)	1.11 (1.02–1.22)
Räisänen S [36]	2012	26,216	1783	633	43	1.00 (0.73–1.37)	--
Emergent CD							
Furuya S [25]	2021	851	174	55	23	2.80 (1.60–4.90)	--
Yang M [29]	2020	59,475	5293	501	81	1.97 (1.55–2.51)	--
Suzuki S [34]	2015	16,797	1439	168	29	2.23 (1.49–3.33)	--
Ebbing C [35]	2013	613,978	48,830	9500	1346	1.91 (1.80–2.03)	1.80 (1.69–1.91)
Räisänen S [36]	2012	26,216	2281	633	97	1.90 (1.52–2.37)	--
Instrumental delivery							
Furuya S [25]	2021	851	158	55	8	0.75 (0.35–1.61)	--
O’Quinn C [7]	2020	30,953	4871	107	16	0.94 (0.55–1.60)	--
Ebbing C [35]	2013	613,978	50,778	9500	783	1.00 (0.93–1.07)	0.90 (0.83–0.97)
Räisänen S [36]	2012	26,216	2124	633	56	1.10 (0.83–1.45)	--

ORs were calculated using RevMan version 5.4.1, and some values were inferred by the authors. Thus, certain values in the table were slightly different from the original values. Abbreviations: CD, cesarean delivery; Control No., number of women in the control group; Control events, number of events in the control group; VCI No., number of women with velamentous cord insertion; VCI events, number of events in the velamentous cord insertion group; Crude OR, crude odds ratio; Adjusted OR, adjusted odds ratio; CI, confidence interval; --, not applicable.

**Table 4 biomedicines-10-01722-t004:** Effect of VCI on the rate of abnormal placentation and malpresentation.

Author	Year	Cont No.	Cont Events	VCI No.	VCI Events	Crude OR (95% CI)	Adjusted OR (95% CI)
Previa							
Yang M [29]	2020	59,475	357	501	8	2.69 (1.33–5.45)	4.58 (3.34–6.27)
Suzuki S [34]	2015	16,797	164	168	5	3.11 (1.26–7.68)	3.24 (0.91–5.5)
Ebbing C [35]	2013	613,978	1713	9500	102	3.88 (3.17–4.74)	3.71 (3.03–4.55)
Räisänen S [36]	2012	26,216	184	633	11	2.50 (1.35–4.62)	2.24 (1.17–4.27)
PASD							
Furuya S [25]	2021	851	16	55	4	4.09 (1.32–12.69)	--
Suzuki S [34]	2015	16,797	97	168	3	3.13 (0.98–9.98)	--

We calculated ORs using RevMan version 5.4.1, and some values were inferred by the authors; thus, certain values in the table were slightly different from the original values. Abbreviations: Cont No., number of women in the control group; Cont events, number of events in the control group; VCI No., number of women with velamentous cord insertion; VCI events, number of events in the velamentous cord insertion group; Crude OR, crude odds ratio; Adjusted OR, adjusted odds ratio; CI, confidence interval; PASD, placenta accreta spectrum of disorder; --, not applicable.

**Table 5 biomedicines-10-01722-t005:** Effect of VCI on the rate of adverse obstetric outcomes.

Author	Year	Cont No.	VCI No.	Cont Age	VCI Age	Cont Nuli	VCI Nuli	Control mal	Vci mal	Control Prior CD	VCI Prior CD
Furuya S [25]	2021	851	55	35.9 ± 3.6	35.8 ± 3.9	596	42	--	--	--	--
Yang M [29]	2020	59,475	501	31.82 ± 4.56	31.73 ± 3.85	23,968	238	--	--	4818	36
O’Quinn C [7]	2020	30,953	107	31.4 ± 4.7	32.2 ± 4.8	13,173	54	--	--	--	--
Kalafat E * [32]	2018	277	50	33 ± 1.06 ^C^	34.3 ± 4.73 ^C^	--	--	--	--	--	--
Ebbing C ^†^ [33]	2017	860,464	13,011	--	--	356,447	5950	--	--	--	--
Suzuki S [34]	2015	16,797	168	--	--	8655	114	--	--	--	--
Ebbing C ^†^ [35]	2013	613,978	9500	--	--	256,136	4292	23,382	759	^A^	^B^
Räisänen S [36]	2012	26,216	633	29.1 ± 5.6	29.3 ± 5.7	10,591	299	--	--	43	2045

* All cases were twin pregnancies. ^†^ Used the same database. We calculated ORs using RevMan version 5.4.1 (Cochrane Collaboration, Copenhagen, Denmark), and some values were inferred by the authors, meaning that they may be different from their original state. A, 51,123/36,7342; B, 802/5208; C, converted median values to mean values with standard deviation using the statistical algorithms reported by Hozo et al. [40]. Abbreviations: No., number of included cases; VCI, velamentous cord insertion; Cont, control; Cont Nuli. The number of nulliparous women in the control group; VCI Nuli, the number of nulliparous women in the velamentous cord insertion group; Control mal, the number of malpresentations in the control group; VCI mal, the number of malpresentations in the velamentous cord insertion group; --, not applicable.

## Data Availability

All the studies used in this study are published in the literature.

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
