# Peer review of "Assisted Reproductive Technique and Abnormal Cord Insertion: A Systematic Review and Meta-Analysis"

_biomedicines, 2022, doi:10.3390/biomedicines10071722_

Round 1
Reviewer 1 Report
Journal: Biomedicines
Type of manuscript: Systematic Review
Title: Association between assisted reproductive technique and ab-normal cord insertion: A systematic review and meta-analysis
Manuscript ID: biomedicines-1546372
This study aimed to examine the relationship between assisted reproductive technology (ART) and abnormal cord insertion (ACI).
Comments and Suggestions for Authors:
The manuscript is an interesting study, but requires some considerations.
1. Introduction:
Acronyms are not used consistently in the manuscript. "Abnormal cord insertion" without using the acronym "ACI" appears repeatedly in the manuscript.
This should be corrected throughout the manuscript.
2. Material and methods:
Page 3, Line 91. “Studies were screened by inspecting the titles and abstracts of applicable studies, as previously reported [21-23]”. The authors should consider whether these references (from the authors themselves) are necessary, as they are different topics.
Page 3, Line 104. Exclusion criteria. ACI has been considered to be increased in the twin and multiple pregnancies. Why weren't they excluded from the review?
Page 3, Line 124. “The risk of bias in the included studies was assessed using the Risk of Bias in Non-randomized Studies-of Interventions tool, as previously performed [24-27]”. The authors should consider whether reference 27 (from the authors themselves) is necessary, as it is different topic.
3. Results:
Page 14, Line 383. Table 5.
"* All cases were twin pregnancies". Why weren't they excluded from the review?
"# All embryos were frozen and transferred on day 5 (blastocyst stage)". It is not indicated to which author this call corresponds.
4. Discussion and 5. Conclusions:
The authors make a great effort to obtain the results of the study and honestly expose its many limitations, especially the heterogeneity of the studies used, with the consequent considerable bias and difficulty in homogenizing the groups and obtaining valid conclusions. Subsection 4.2. Strengths and limitations, in its reference to limitations, is the most interesting part of the study.
The primary objective (relationship between ART and ACI) is set forth in Abstract, Page 1, Line 24. “In conclusion, ART may be correlated with an increased prevalence of ACI” and Conclusions, Page 20, Line 583. “In the current goal-analysis, ART was associated with an increased rate of ACI”. However, Discussion, Page 16, Line 449. It is commented in subsection 4.2. Strengths and limitations: “Women conceived by ART may be more likely to be of advanced maternal age and have multiple gestation; thus, excluding confounding factors is essential. However, most studies [7,33,34,36,38,40,43] did not perform a multivariate analysis with adjustments for patient background; thus, our analysis cannot characterize ART as a risk factor for ACI”. This contradiction must be explained and in the Abstract and Conclusions sections the limitations that make the results of the study must be considered with extreme caution must be clearly exposed.
6. References:
References should be thoroughly revised to conform to uniform and appropriate standards for the journal Biomedicines.
Author Response
The authors would like to thank the reviewers for their constructive critique to improve the manuscript. We have made every effort to address the issues raised and to respond to all comments. The revisions made in the manuscript are indicated using the “track changes” function of Microsoft Word. Please, find next a detailed, point-by-point response to the comments. We hope that our revisions would meet the reviewers’ expectations.
Reviewer #1
Journal: Biomedicines
Type of manuscript: Systematic Review
Title: Association between assisted reproductive technique and ab-normal cord insertion: A systematic review and meta-analysis
Manuscript ID: biomedicines-1546372
This study aimed to examine the relationship between assisted reproductive technology (ART) and abnormal cord insertion (ACI).
Comments and Suggestions for Authors:
The manuscript is an interesting study, but requires some considerations.
Response:
The authors would like to thank the reviewer for his/her constructive critique to improve the manuscript. We have made every effort to address the issues raised and to respond to all comments. Please, find next a detailed, point-by-point response to the reviewer's comments. We hope that our revisions would meet the reviewer’s expectations.
Reviewer #1, comment #1
- Introduction:
Acronyms are not used consistently in the manuscript. "Abnormal cord insertion" without using the acronym "ACI" appears repeatedly in the manuscript.
This should be corrected throughout the manuscript.
Reply: Throughout the manuscript
We would like to thank the reviewer for the helpful comments. According to the reviewers’ suggestion, we have used the acronym “ACI” throughout the manuscript.
Reviewer #1, comment #2
- Material and methods:
Page 3, Line 91. “Studies were screened by inspecting the titles and abstracts of applicable studies, as previously reported [21-23]”. The authors should consider whether these references (from the authors themselves) are necessary, as they are different topics.
Reply: Line 101
We would like to thank the reviewer for the comments. As the reviewer suggested, we have removed the unnecessary citations.
Reviewer #1, comment #3
Page 3, Line 104. Exclusion criteria. ACI has been considered to be increased in the twin and multiple pregnancies. Why weren't they excluded from the review?
Reply: Line 130, lines 222-225, lines 262-265, lines 412-415
We would like to thank the reviewer for the insightful comments. We completely agree with the reviewer’s opinion; however, the exclusion criteria were defined in the protocol, and we could not change them in this phase. To aid this problem, we have performed a sensitivity analysis, in which studies that included only women with twin pregnancies were excluded. The results of sensitivity analysis are presented in the revised manuscript as follows:
“In the sensitivity analysis, studies that included only women with twin pregnancy [32,37] were excluded. In this unadjusted pooled analysis (n=8) [7,29-31,33,34,36,39], women with ART pregnancy were more likely to have VCI compared to those with non-ART pregnancy (OR 2.28, 95%CI, 1.64–3.15; heterogeneity, P <0.01, and I²=77%).” (Lines 222–225)
“In the sensitivity analysis, studies with only twin pregnancies [32,37] were excluded. In the unadjusted pooled analysis, (n=4) [7,30,33,39], ART pregnancies were correlated with a higher MCI rate compared to non-ART pregnancies (OR 1.74, 95%CI 1.49–2.04; heterogeneity, P<0.01; I²=42%).” (Lines 262–265)
“In the sensitivity analysis, the study that included only twin pregnancy was removed [32], and fixed effect analysis was performed. The results of sensitivity analysis were similar with those of primary analysis (n=4; mean difference, 0.07; 95%CI, -0.18–0.32; heterogeneity, P=0.59; and I²=20%).” (Lines 412–415)
Reviewer #1, comment #4
Page 3, Line 124. “The risk of bias in the included studies was assessed using the Risk of Bias in Non-randomized Studies-of Interventions tool, as previously performed [24-27]”. The authors should consider whether reference 27 (from the authors themselves) is necessary, as it is different topic.
Reply: Line 135
We would like to thank the reviewer for the comments. As per the reviewer’s suggestion, we have removed the unnecessary citations.
Reviewer #1, comment #5
- Results:
Page 14, Line 383. Table 5.
"* All cases were twin pregnancies". Why weren't they excluded from the review?
Reply: Line 130, lines 222-225, lines 262-265, lines 412-415
Please also refer to the reply to the Reviewer #1, comment #3. To meet the reviewer’s suggestion, we have performed sensitivity analyses after excluding the studies that included only women with twin pregnancy. The results of sensitivity analyses are presented in the revised manuscript as follows:
“In the sensitivity analysis, studies that included only women with twin pregnancy [32,37] were excluded. In this unadjusted pooled analysis (n=8) [7,29-31,33,34,36,39], women with ART pregnancy were more likely to have VCI compared to those with non-ART pregnancy (OR 2.28, 95%CI, 1.64–3.15; heterogeneity, P <0.01, and I²=77%).” (Lines 222–225)
“In the sensitivity analysis, studies with only twin pregnancies [32,37] were excluded. In the unadjusted pooled analysis, (n=4) [7,30,33,39], ART pregnancies were correlated with a higher MCI rate compared to non-ART pregnancies (OR 1.74, 95%CI 1.49–2.04; heterogeneity, P<0.01; I²=42%).” (Lines 262–265)
“In the sensitivity analysis, the study that included only twin pregnancy was removed [32], and fixed effect analysis was performed. The results of sensitivity analysis were similar with those of primary analysis (n=4; mean difference, 0.07; 95%CI, -0.18–0.32; heterogeneity, P=0.59; and I²=20%).” (Lines 412–415)
Reviewer #1, comment #6
"# All embryos were frozen and transferred on day 5 (blastocyst stage)". It is not indicated to which author this call corresponds.
Reply: Table 1
As per the reviewer’s insightful suggestion, we have revised Table 1.
Reviewer #1, comment #7
- Discussion and 5. Conclusions:
The authors make a great effort to obtain the results of the study and honestly expose its many limitations, especially the heterogeneity of the studies used, with the consequent considerable bias and difficulty in homogenizing the groups and obtaining valid conclusions. Subsection 4.2. Strengths and limitations, in its reference to limitations, is the most interesting part of the study.
The primary objective (relationship between ART and ACI) is set forth in Abstract, Page 1, Line 24. “In conclusion, ART may be correlated with an increased prevalence of ACI” and Conclusions, Page 20, Line 583. “In the current goal-analysis, ART was associated with an increased rate of ACI”. However, Discussion, Page 16, Line 449. It is commented in subsection 4.2. Strengths and limitations: “Women conceived by ART may be more likely to be of advanced maternal age and have multiple gestation; thus, excluding confounding factors is essential. However, most studies [7,33,34,36,38,40,43] did not perform a multivariate analysis with adjustments for patient background; THUS, OUR ANALYSIS CANNOT CHARACTERIZE ART AS A RISK FACTOR FOR ACI”. This contradiction must be explained and in the Abstract and Conclusions sections the limitations that make the results of the study must be considered with extreme caution must be clearly exposed.
Reply: Abstract, lines 686-688
We appreciate the reviewer’s comments and we completely agree with his/her opinion. According to the reviewer’s suggestion, we have presented the limitations of this study in the Abstract and Conclusion sections as follows:
“However, most studies could not exclude confounding factors; thus, further studies are warranted to characterize ART as a risk factor for ACI. In women with ACI, elective and emergent CD rates are high.” (Lines 24–26)
“In the current meta-analysis, ART may be associated with an increased rate of ACI, and ACI was correlated with an increased rate of elective and emergent CD. As the number of included studies is still limited and since most previous studies could not exclude confounding factors, extreme caution is needed when the results of our study are interrupt.” (Lines 686–688)
Reviewer #1, comment #8
- References:
References should be thoroughly revised to conform to uniform and appropriate standards for the journal Biomedicines.
Reply: References
Please note that we have formatted the references as per the journal guidelines.

Reviewer 2 Report
I suggest the title should be changed to more accurately reflect the findings. It currently implies different ART techniques show different correlations with cord insertion and this is not really true.
line 68: Pleas add how high is the prevalence is.
The aim of the study switches between correlating ART with cord insertion or unprotected cord. Please be more consistent as this could be confusing.
Why was the date May 31, 2021 chosen? How many studies have there been since this time?
Please clarify what you are defining as 'events' in table 1.
The statements of the secondary outcomes would be more powerful if they stated the result of the outcome rather than what was looked at.
Subheadings in the discussion section do not appear to be consistent with the journal format of other published manuscripts for this publication.
Can you please explain (line 509) why LLP to be higher in ART when as you state there is no data.
Author Response
The authors would like to thank the reviewer for his/her constructive critique to improve the manuscript. We have made every effort to address the issues raised and to respond to all comments. Please, find next a detailed, point-by-point response to the reviewer's comments. We hope that our revisions would meet the reviewer’s expectations.
Reviewer #2, comment #1
I suggest the title should be changed to more accurately reflect the findings. It currently implies different ART techniques show different correlations with cord insertion and this is not really true.
Reply: Title
We appreciate the reviewer’s insightful comments. According to the reviewer’s suggestion, we have revised the title of the manuscript as follows:
“Assisted reproductive technique and abnormal cord insertion: A systematic review and meta-analysis.”
Reviewer #2, comment #2
line 68: Pleas add how high is the prevalence is.
Response: Lines 73-74
Please note that we have clarified the prevalence of vasa previa in women who conceived by assisted reproductive technology. The added part is as follows:
“Previous studies have found that the prevalence of vasa previa appears to be high (approximately 0.3–0.5%) in women who conceived using ART [9,16-19]; however, comparative studies examining the rate of vasa previa between ART pregnancy and spontaneous pregnancy are scarce.” (Lines 73–74)
Reviewer #2, comment #3
The aim of the study switches between correlating ART with cord insertion or unprotected cord. Please be more consistent as this could be confusing.
Reply: Line 79, line 123
As the reviewer suggested, the aim of this study was confusing. To address this concern, we have revised the aim of this study from “Since an unprotected cord may be associated with adverse delivery outcomes, the relationship between ACI and delivery outcomes was also examined.” To “As an ACI may be associated with adverse delivery outcomes, the relationship between ACI and delivery outcomes was also examined.” (Lines 79–80)
Reviewer #2, comment #4
Why was the date May 31, 2021 chosen? How many studies have there been since this time?
Reply: Lines 95-96, lines 548-555
We would like to thank the reviewer for the questions. This study was registered on June, 2021 in PROSPERO, and the search date was defined in the protocol as follows:
“We will conduct a systematic search of articles published before May 31, 2021, using the PubMed, Scopus, and Cochrane Central Register of Controlled Trials (CENTRAL) databases.”
Therefore, the end of search was on May 31, 2021. We have provided this information in the revised manuscript as follows:
“Literature published before May 31, 2021, was screened using words related to ART and ACI. The used search engines and search date were based on the pre-registered protocol in PROSPERO. Medical subject headings (MeSH terms) were used in the PubMed and Cochrane database searches.” (Lines 85–88)
After the end of search date, we identified two studies that met the inclusion criteria of this study (PMID: 34787339 and 34077551). Since the present study was a pre-registered systematic review, we could not include these studies. Nevertheless, we cited and discussed an important study that showed the increased rate of vasa previa in women with ART compared to those without ART (PMID: 34077551).
Reviewer #2, comment #5
Please clarify what you are defining as 'events' in table 1.
Reply: Table 1
As per the reviewer’s insightful suggestion, we have clarified the definition of events mentioned in Table 1.
Reviewer #2, comment #6
The statements of the secondary outcomes would be more powerful if they stated the result of the outcome rather than what was looked at.
Reply: Line 320, line 354, line 366
We would like to thank the reviewer for the valuable suggestion. According to the reviewer’s suggestion, we have revised the statements of the secondary outcomes.
Reviewer #2, comment #7
Subheadings in the discussion section do not appear to be consistent with the journal format of other published manuscripts for this publication.
Reply: Lines 630-671
We would like to thank the reviewer for the helpful comments. As the reviewer suggested the “Strength and limitations” subsection has been moved before the Conclusion section.
Reviewer #2, comment #8
Can you please explain (line 509) why LLP to be higher in ART when as you state there is no data.
Reply: Lines 561-563, line 565
We sincerely appreciate the reviewer’s suggestion. Please note that we re-evaluated the studies that were included in the previous systematic review and found that one study showed the increased incidence of low-lying placenta in ART pregnancy compared to those in non-ART pregnancy. Therefore, we have revised the statement. Nevertheless, as the reviewer suggested, the expression of our statement regarding relationship between ART and low-lying placenta was too strong; thus, we have weakened the expression of the statement. The revised part is as follows:
“Nevertheless, only one study showed a significantly higher incidence of low-lying placenta in ART pregnancy compared to that in non-ART pregnancy (adjusted OR, 1.83; 95% CI, 1.04–3.12) [55].
Although specific data regarding low-lying placenta is scant, we hypothesized that both women with VCI and women who conceived by ART has a potential to have a higher rate of low-lying placenta than women who conceived normally (Figure 7b). Based on the results of these meta-analyses, a possible mechanism for the increased rate of type I vasa previa in ART pregnancies is presented in Figure 7.” (Lines 561–569).

Round 2
Reviewer 1 Report
Journal: Biomedicines
Type of manuscript: Systematic Review
Title: Association between assisted reproductive technique and abnormal cord insertion: A systematic review and meta-analysis
Manuscript ID: biomedicines-1546372
Round 2:
The authors make changes in the new version that clarify the manuscript.
Author Response
Reviewer #1
Journal: Biomedicines
Type of manuscript: Systematic Review
Title: Association between assisted reproductive technique and abnormal cord insertion: A systematic review and meta-analysis
Manuscript ID: biomedicines-1546372
Round 2:
The authors make changes in the new version that clarify the manuscript.
Reply:
Thank you for reviewing our paper and considering it for publication in Biomedicines. We have reviewed our manuscript carefully and some minor mistakes were corrected. We trust that the revised manuscript will now be suitable for publication as an Original Research Article in Biomedicines.
